# The Methanol Extract of *Allium cepa* L. Protects Inflammatory Markers in LPS-Induced BV-2 Microglial Cells and Upregulates the Antiapoptotic Gene and Antioxidant Enzymes in N27-A Cells

**DOI:** 10.3390/antiox8090348

**Published:** 2019-09-01

**Authors:** Md. Jakaria, Shofiul Azam, Duk-Yeon Cho, Md. Ezazul Haque, In-Su Kim, Dong-Kug Choi

**Affiliations:** 1Department of Applied Life Sciences and Integrated Bioscience, Graduate School, Konkuk University, Chungju 27478, Korea; 2Department of Integrated Bioscience and Biotechnology, College of Biomedical and Health Sciences, and Research Institute of Inflammatory Diseases (RID), Konkuk University, Chungju 27478, Korea

**Keywords:** *Allium cepa*, protection, neurodegenerative diseases, inflammatory markers, Bcl-2, antioxidant enzymes

## Abstract

Neuroinflammation, apoptosis, and oxidative stress are connected to the pathogenesis of neurodegenerative diseases (NDDs). Targeting these three factors, the intervention of neuroprotective agents may have great potential in the treatment of NDDs. In the current study, the anti-inflammatory effects of the methanol extract of *Allium cepa* (MEAC) in lipopolysaccharide (LPS)-induced BV-2 microglial cells were investigated. MEAC has been studied in regard to the regulation of the antiapoptotic gene (Bcl-2) and various antioxidant enzyme (HO-1, NQO-1, and catalase) expressions in N27-A cells. Additionally, the protective action of MEAC has also been studied against MPP^+^-induced death in N27-A cells. The results suggest that MEAC is significantly protected from NO release and increase iNOS expression at the mRNA and protein levels in LPS-stimulated BV-2 microglial cells. MEAC treatment also protects COX-2 expression at the mRNA and protein levels. Furthermore, MEAC treatment prevents LPS-stimulated increases of proinflammatory cytokines, including TNF-α, IL-6, and IL-1β. In N27-A cells, MEAC treatment significantly upregulates antiapoptotic gene (Bcl-2) and antioxidant enzyme (HO-1, NQO1, and catalase) expressions. Moreover, MEAC treatment protects against MPP^+^-induced death in N27-A cells. To conclude, *A cepa* extract takes protective action against LPS and MPP^+^, and upregulates the antioxidant enzymes that could potentially be used in the therapy of NDDs.

## 1. Introduction

Progressive dysfunction and toxicities within neurons characterize the debilitating multifactorial disorders known as neurodegenerative diseases (NDDs). These diseases lead to a slow but irreversible weakening of all brain function. Around 30 million individuals worldwide are affected by NDDs. Although some symptomatic treatments are available, specific treatments have not yet been discovered [1]. Neurons and microglial cells are considered to be key parts of the central nervous system (CNS), and microglial cells occupy 5–10% of brain cells [2]. In a healthy brain, microglia play a vital role in neuroprotection, and phagocytes remove cell debris and damaged neurons [3,4]. On the other hand, the neuroinflammatory and neurotoxic responses are accelerated from activated microglia and astrocytes by releasing numerous inflammatory markers, includingcyclooxygenase-2 (COX-2), nitric oxide synthase (iNOS), interleukin-6 (IL-6), tumor necrosis factor-α (TNF-α), and interleukin-1β (IL-1β) [2,5,6].

The deregulation of apoptosis and neuroinflammatory responses is strongly correlated with NDDs [7,8]. Both apoptosis and neuroinflammation are triggered by oxidative stress (an imbalance between the generation of free radicals and antioxidant defenses) [9,10,11]. The regulation of oxidative stress, neuroinflammation, and apoptosis represent an emerging approach for protecting neurodegeneration-related disease.

A bulbous herb, *Allium cepa* L. (onion), belongs to the Alliaceae family and is cultivated worldwide for commercial purpose [12]. The dietary intake of onions is connected to a reduced risk of developing different types of cancer, cardiovascular disease, and NDDs as per several epidemiological studies [13]. The presence of high content of phytoactive constituents, such as phenolic compounds, particularly flavonoids and numerous organosulfur compounds, are attributed to the beneficial effect of onions [13]. Several studies have noted the ameliorating effect of *A. cepa* (AC) against neurological disorders. *A. cepa* extract protects against ischemia and reperfusion-induced cerebral injury, and aluminum-induced neurotoxicity in rodent models [14,15].

A recent study explores how the methanol extracts of *A. cepa* and quercetin prevent cortical neuronal cells from oxidative stress via protein kinase c-ε inactivation/extracellular signal-regulated kinase-1/2 activation [16]. The current study aims to examine the preventive action of the methanol extract of *A. cepa* (MEAC) against lipopolysaccharide (LPS)-induced inflammatory markers in BV-2 microglial cells. This study also analyzes the regulatory effect of MEAC on an antiapoptotic gene, B-cell lymphoma 2 (Bcl-2), and several antioxidant enzymes such as hemeoxygenase-1 (HO-1), NAD(P)H Quinone Dehydrogenase 1 (NQO1), and catalase in dopaminergic N27-A cells. Finally, the preventive action of MEAC is investigated against 1-methyl-4-phenylpyridinium (MPP^+^)-induced death in N27-A cells.

## 2. Materials and Methods 

### 2.1. Materials

MEAC (Korea Plant Extract Bank, Ref no: PB2051.2), LPS (Escherichia coli; 055:B5), MPP^+^, dimethyl sulfoxide (DMSO), N-(1-naphthyl)ethylenediamine dihydrochloride, sulfanilamide, 3-(4,5-dimethylthiazol-2-yl)-2,5-diphenyltetrazolium bromide (MTT), and sodium nitrite were obtained from Sigma-Aldrich (St. Louis, MO, USA). Fetal bovine serum (FBS) and phosphate-buffered saline were procured from Gibco/Invitrogen (Carlsbad, CA, USA). The RPMI was obtained from Corning (USA), and Dulbecco’s modified Eagle medium (DMEM) was obtained from Gibco/Invitrogen (Carlsbad, CA, USA). The protease and phosphatase inhibitors were obtained from Roche (Indianapolis, IN, USA), and the cell culture plates were procured from Nunc Inc. (Aurora, IL, USA).

### 2.2. BV-2 and N27-A Cell Cultures

The obtaining of BV-2 microglial cells was noted previously [17,18]. The cells were cultured in DMEM and supplemented with 5% FBS and 1% of 100 units/mL of penicillin/streptomycin. The rat dopaminergic cell line, N27-A, was obtained from the Gates Center for Stem Cell Research and Regenerative Medicine, and the Human Medical Genetics and Genomics Program, University of Colorado School of Medicine, Aurora, Colorado, USA. The N27-A cells were cultured in an RPMI 1640 medium (10% FBS, 2 mM L-glutamine, and 100 U/mL of penicillin and streptomycin). Both cell lines were maintained at 37 °C in a 5% CO_2_ and 95% humidified air incubator for the indicated time cells. MEAC was dissolved in DMSO. In all of the experiments, cells were seeded at a density of 2.5 × 10^5^ cells/mL.

The expressions of inflammatory markers were investigated according to protocol of Cho et al. [2] with slight modifications. BV-2 microglial cells were pretreated with MEAC (50 μg/mL, 250 μg/mL, and 500 μg/mL). The LPS (200 ng/mL) solution was treated 1 h later than the MEAC treatment. The treated cells were then incubated for the indicated times (6 h for mRNA and 18 h protein). N27-A cells were treated with MEAC (500 μg/mL) for 18 h to check the regulatory action of the antiapoptotic gene (Bcl-2) and antioxidant enzymes (HO-1, NQO1, and catalase). All of the experiments were done for three individual sets.

### 2.3. Cell Viability and Nitrite Assay

The BV-2 microglial cells were seeded at a density of 2.5 × 10^5^ cells/mL and were co-treated with various concentrations of MEAC (50 μg/mL, 250 μg/mL, and 500 μg/mL) and LPS (200 ng/mL) for 24 h. The viability of the cells was measured according to the method of Park et al. [19]. The nitrite assay procedure was followed according to the method of Cho et al. [2]. The BV-2 microglial cells were seeded at a density of 2.5 × 10^5^ cells/mL and then treated with various concentrations of MEAC (50 μg/mL, 250 μg/mL, and 500 μg/mL) and LPS (200 ng/mL) for 24 h for the nitrite assay. To check the protective action of MEAC against MPP^+^-induction, the N27-A cells were pretreated with several concentrations of MEAC and, 2 h later, MPP^+^ was treated and harvested for 24 h and cell viability was calculated by following the protocol of Park et al. [19].

### 2.4. Isolation of Total RNA and Reverse Transcription Polymerase Chain Reaction (RT-PCR)

A trizol reagent (Invitrogen Life Technologies, Carlsbad, CA, USA) was used to isolate the total RNA from the BV-2 microglial cells and N27-A cells following the manufacturer’s guidelines. A first-strand cDNA synthesis was performed using 2.5 μg of total RNA with the ReverTra Ace-α kit (Toyobo, Osaka, Japan). The reaction was executed at 60°C for 60 min and was then heated at 95 °C for five min; 1 μL from each RT-reaction mixture was used for PCR amplification. PCR amplification was accomplished using specific primers (Bioneer, Daejeon, Korea), as reported in Table 1. By using 1.2% agarose-gel electrophoresis, all of the PCR products were resolved and visualized with the GelRed® Nucleic Acid Gel Stain (Biotium, Inc., Landing Parkway, Fremont, CA, USA). The gels were photographed; the pixel intensity for each band was determined in Image J (NIH), and they were normalized to the band intensity of GAPDH mRNA.

### 2.5. Western Blot Analysis

The cells were placed at 4 °C upon the 2-time washing with PBS and lysed for 10 min in a lysis buffer (1× RIPA lysis buffer, protease-inhibitor cocktail, and phosphatase-inhibitor cocktail). The lysates were centrifuged at 14,000 rpm at 4 °C, and the collected supernatants were collected for further investigation. Equal amounts of protein (20 μg or 40 μg) were separated electrophoretically using a 10% sodium dodecyl sulfate-polyacrylamide electrophoresis, and the resolved proteins were transferred to polyvinylidene-difluoride membranes (Millipore, Bedford, MA, USA). The membranes were incubated for 1 h with 3% bovine serum albumin in Tris-buffered saline; a 0.1% Tween 20 buffer was used to prevent nonspecific binding.

Subsequently, the blots were incubated with primary antibodies, including anti-iNOS (1:2000; Cell Signaling Technology, Danvers, MA, USA), anti-COX-2 (1:2000; Santa Cruz Biotechnology, Santa Cruz, CA, USA), and anti-β-actin (1:5000; Santa Cruz Biotechnology, Santa Cruz, CA, USA). The blots were visualized with an enhanced chemiluminescence detection system (Santa Cruz Biotechnology) as per the recommended protocol.

### 2.6. Statistical Analysis

A Graph Pad Prism ver. 5.01 (Graph Pad, Inc., La Jolla, CA, USA) was employed to analyze experimental data. All of the data is expressed as the mean ± standard error of at least three independent experiments that were performed in triplicate. A one-way variance analysis and Tukey’s multiple-comparison test were employed for the statistical analysis; *p* < 0.05 is considered statistically significant.

## 3. Results

### 3.1. Action of MEAC Treatment on LPS-Induced Cell Viability and NO Production

Several concentrations of MEAC (50 μg/mL, 250 μg/mL and 500 μg/mL), and/or LPS (200 ng/mL) were primarily treated to BV-2 microglial cells to check whether they exert any toxicity in MTT assay. MEAC (50 μg/mL, 250 μg/mL, and 500 μg/mL) treatments with LPS (200 ng/mL) did not affect the BV-2 microglial cells viability (Figure 1B).

MEAC action on LPS-induced nitric oxide (NO) production was assessed by the Griess assay. Cotreatment of MEAC (50 μg/mL, 250 μg/mL and 500 μg/mL) concentrations and LPS (200 ng/mL) were treated to microglial cells for 24 h. LPS induction markedly elevated NO production (18.8 ± 0.55 μM) compared to the control (2.52 ± 0.12 μM). As per results, the treatment with MEAC notably prevented the NO production in a concentration-dependent manner to 14.12 ± 0.21 μM, 10.71 ± 0.10 μM, and 6.03 ± 0.04 μM, respectively, in the LPS-stimulated BV-2 cells (Figure 1A).

### 3.2. Action of MEAC on LPS-Induced iNOS Expressions in BV-2 Microglial Cells at mRNA and Protein Levels

As the MEAC treatment reduced the NO concentration, iNOS-mRNA and -protein expressions were evaluated further. The stimulation (200 ng/mL) of LPS noticeably upregulated the iNOS-mRNA level to 4.23 ± 0.18 folds after 6 h in comparison with the control group (Figure 2A). In a concentration-dependent manner, MEAC pretreatment prevented the iNOS-mRNA in the LPS-induced BV-2 microglial cells. The western blot analysis evaluated the iNOS expression at the protein level. At 18 h after the LPS stimulation, the iNOS-protein level was noticeably elevated to 4.95 ± 0.056 folds compared to the control group. Meanwhile, the MEAC treatment significantly prevented iNOS expression at the protein level (Figure 2B).

### 3.3. Action of MEAC on LPS-Induced COX-2 Expressions in BV-2 Microglial Cells at mRNA and Protein Levels

BV-2 microglial cells were treated with LPS in the absence or presence of MEAC for 6 h or 18 h to investigate COX-2 expression at the mRNA and protein levels. Stimulation with LPS for 6 h significantly elevated the COX-2 gene expression compared with that of the control group; however, the pretreatment with MEAC 1 h before LPS stimulation significantly prevented the COX-2-mRNA expression in a dose-dependent manner (Figure 3A). In addition, LPS treatment for 18 h also dose-dependently increased COX-2 expression at the protein level, while MEAC pretreatment prevented LPS-induced increase COX-2 expression (Figure 3B).

### 3.4. Action of MEAC on LPS-Induced Expressions of Inflammatory Cytokines in BV-2 Microglial Cells at mRNA Level

MEAC treatments suppressed the LPS-induced expression of IL-6, TNF-α, and IL-1β in BV-2 microglial cells as per the RT-PCR analysis. The induction of LPS significantly elevated proinflammatory cytokine (IL-6, TNF-α, and IL-1β) expressions, while the pretreatment with MEAC 1 h before LPS treatment prevented an increase in expression of those markers (Figure 4A–D).

### 3.5. Protective Activity of MEAC Against MPP^+^-Induced N27-A Cells

Various concentrations of MEAC (25, 50, 100, 250, and 500 µg/mL) were treated in N27-A cells to investigate whether the treatment causes mortality. The results indicate that being treated with MEAC for 24 h does not cause any significant cytotoxicity in N27-A cells (Figure 5A). Indeed, the MEAC treatment (500 µg/mL) significantly protects N27-A cells from MPP^+^-induced cell death (Figure 5B).

### 3.6. Regulatory Effect of MEAC on Bcl-2, HO-1, NQO1, and Catalase Expression in N27-A Cells

The activity of MEAC treatment on the regulation of mRNA expressions of Bcl-2, HO-1, NQO1, and catalase is displayed in Figure 6A. To investigate the regulatory action of MEAC on Bcl-2 expression, N27-A cells were treated with MEAC (500 μg/mL) for 18 h. The treatment significantly upregulated Bcl-2 expression to 1.48 ± 10 folds compared to the control group (Figure 6B). N27-A cells were treated by MEAC (500 μg/mL) to evaluate the regulatory effect on HO-1 and NQO1 expressions. The treatment with MEAC for 18 h significantly upregulated expression of HO-1 to 1.85 ± 0.04 folds compared to the control group (Figure 6C). In comparison with the control group, MEAC extract significantly upregulated NQO1 expression to 2.72 ± 0.25 (Figure 6D). Apart from HO-1 and NQO1, N27-A cells were treated with MEAC to check its regulatory role on catalase expression. An 18-h MEAC (500 μg/mL) treatment significantly upregulated catalase expression to 1.78 ± 0.04 folds in comparison with the control group (Figure 6E).

## 4. Discussion 

The elucidation of the mechanism of progression and therapeutics of NDDs remains a significant challenge [20]. Recently, however, several natural products produced protective actions in the model of NDDs [21,22,23,24,25]. The natural product-derived intervention of therapeutics could have potential in the therapy of NDDs [21].

The design of studies focused on toxic agent-intoxicated neuronal injury may aid to find possible therapeutics for treating NDDs [20]. LPS is a well-known cell wall component of gram-negative bacteria, which has the ability to trigger the microglia. LPS-induced activation of microglia leads to inflammatory responses that initiate the progression of disease in models of neurodegeneration. LPS interacts with Toll-like receptors (TLRs), such as TLR4, on the surface of the microglia. LPS and TLR4 interaction activate several downstream signaling cascades. The activation of TLR4 signaling influences NF-κB and/or other transcription factors in the nucleus and triggers the proinflammatory cytokine release [3]. Thus, for preventing/treating neuroinflammation-related diseases, the modulation of LPS and TLR interaction and/or activation has a potential value.

In a mouse MCAO model, AC extract prevents brain edema, hyperpermeability of blood–brain barrier, and disruption of tight junction proteins, probably through its antioxidant effects. AC extract may be a potential nutrient in the prevention of blood–brain barrier function during brain ischemia [26]. This study investigated the actions of MEAC extracts against LPS-stimulated increased expressions of inflammatory markers. According to this study, the treatment with MEAC protected from the LPS-induced NO release in BV-2 microglial cells. In the brain, prostaglandins are crucial when neuroinflammation occurs. The synthesizing enzymes COX-2 and iNOS are some of the most-referred proinflammatory mediators [2]. As per this study, MEAC also prevents the expression of COX-2 and iNOS at the mRNA and protein levels. In addition, in the BV-2 microglial cells, the elevation of the expression of inflammatory cytokines such as IL-6, TNF-α, and IL-1β is a pointer of neuronal cell damage [27]. Various concentrations of MEAC were treated to reveal the action against inflammatory cytokine expressions. The present study has found that pretreatment with MEAC exhibits preventive actions against LPS-stimulated increase mRNA expressions of IL-6, TNF- α, and IL-1β in BV-2 microglial cells. However, further investigations are needed to confirm MEAC action on cytokine expressions at the protein level.

In the progression of NDDs, apoptosis plays a critical role, according to several animal and cellular model studies. However, research on human brains has demonstrated diverse outcomes. In the brains of NDD patients, much-died neurons are detected and these conditions often accompany significant cell loss as a result of distinctive morphological characteristics of apoptosis, including DNA fragmentation, chromatin condensation, and cysteine-proteases, caspases activation. NDDs and cell death are associated with oxidative stress [28,29].

As an antiapoptotic marker against acute neural injury, Bcl-2 belongs to the Bcl-2 family of apoptotic regulators and is vital for cell survival. The downregulation of the Bcl-2 protein often precedes apoptosis. On the contrary, other members of the Bcl-2 family, for example, Bcl-2 antagonist killer 1 (Bak) and Bcl-2 Associated X (Bax) promote apoptosis by increasing the release of cytochrome c from the mitochondria and activating caspases. Bcl-2 can avert the activation of Bax via heterodimerization. For the cell survival, the interplay of antiapoptotic and proapoptotic regulators is crucial, and Bax/Bcl-2 ratio constitutes an important factor of the apoptotic status of the cell. Pro and antiapoptotic members of the Bcl-2 family are extensively expressed throughout the CNS. Thus, the upregulation of Bcl-2 expression may improve neuroprotective activity [30,31,32]. Based on the results, MEAC treatment upregulates Bcl-2 expression.

Several endogenous and exogenous antioxidant enzymes have been found to play a neuroprotective role in the model of NDDs [33]. Nrf2 and its downstream phase II genes (for instance, HO-1 and NQO1) have potent capability to detoxify harmful compounds, combat reactive oxygen species, and directly or indirectly modulate the BBB permeability, immune system, and inflammatory response. Thus, they are considered as the emerging targets for the treatment of neurological diseases. Indeed, various inducers of Nrf2 have been proven effective in familiar neurological disease models [34]. According to the study results, MEAC treatment upregulates the expression of antioxidant enzymes, including HO-1, NQO1, and catalase.

MPP^+^ is one of the agents that induced cytotoxicity in the experimental model; it causes oxidative stress-mediated cell death [35]. The protective role of MEAC against MPP^+^ induction was also studied in this study. The treatment with MEAC for 24 h protects from MPP^+^-induced N27-A cell death. The stability of MEAC in plasma could be studied further based on the data on both cell lines.

## 5. Conclusions

MEAC suppresses the LPS-induced production of proinflammatory mediators and cytokines in BV-2 microglial cells. MEAC extract also upregulates the antiapoptotic gene (Bcl-2) and antioxidant enzyme (HO-1 and NQO1) expressions in N27-A cells. Therefore, our study results support the previously published outcomes of *A. cepa* extract against stroke and metalloid-induced neurotoxicity in rodents. Future research should examine the *A. cepa* action on NDDs model. This study promotes further investigation into the role of *A. cepa* extract against neurodegenerative disease.

## Figures and Tables

**Figure 1 antioxidants-08-00348-f001:**
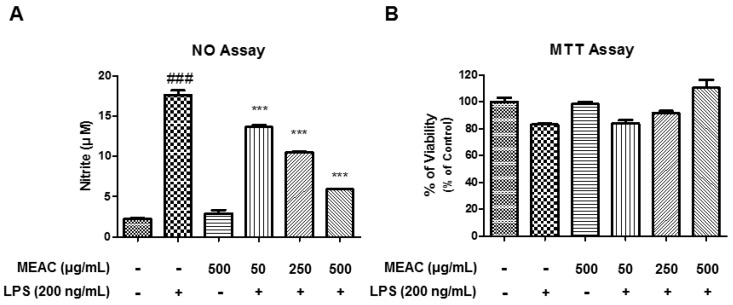
Effect of methanol extract of *A. cepa* (MEAC) on nitric oxide (NO) production and cell viability in lipopolysaccharide (LPS)-induced BV-2 microglial cells. (**A**) The treatment with MEAC on the suppression of LPS-stimulated NO release in BV-2 microglial cells. (**B**) MEAC treatment-mediated cell viability was evaluated using the 3-(4,5-dimethylthiazol-2-yl)-2,5-diphenyltetrazolium bromide (MTT) assay. The results present as a percentage of the control samples. Data are presented as the mean ± SEM. ### *p* < 0.001 versus untreated group; *** *p* < 0.001 versus LPS group (*n* = 3).

**Figure 2 antioxidants-08-00348-f002:**
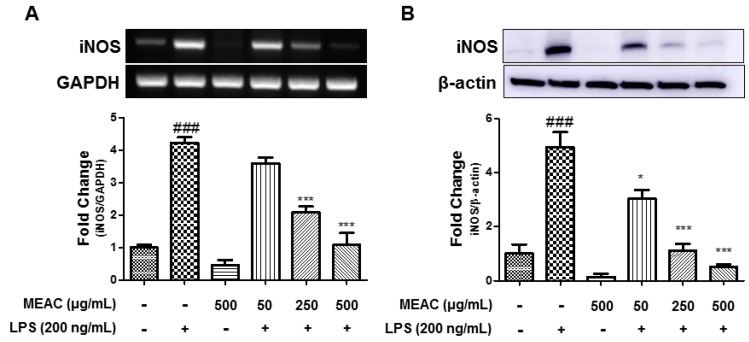
Effect of MEAC on LPS-induced iNOS expressions at mRNA and protein level in BV-2 microglial cells. (**A**) MEAC-treated prevention of iNOS expression at mRNA level. (**B**) The treatment with MEAC protects LPS-stimulated iNOS expression at protein level. Data are presented as the mean ± SEM. ### *p* < 0.001 versus control group (*n* = 3); * *p* < 0.05 and *** *p* < 0.001 versus LPS group (*n* = 3).

**Figure 3 antioxidants-08-00348-f003:**
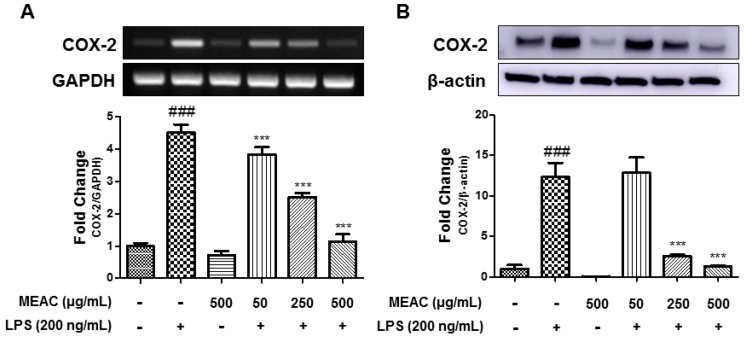
Effect of MEAC on LPS-induced COX-2 expressions at mRNA and protein levels in BV-2 microglial cells. (**A**) MEAC-treated prevention of COX-2 expression at mRNA level. (**B**) The treatment with MEAC protects LPS-stimulated COX-2 expression at protein level. Data are presented as the mean ± SEM. ### *p* < 0.001 versus control group (*n* = 3); *** *p* < 0.001 versus LPS group (*n* = 3).

**Figure 4 antioxidants-08-00348-f004:**
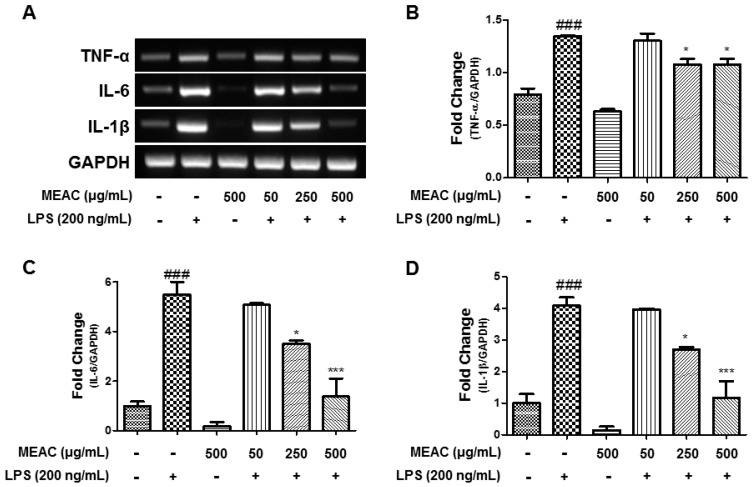
MEAC action on LPS-induced cytokine expressions in BV-2 microglial cells. (**A**) The regulation of TNF-α, IL-1β, and IL-6 mRNA expressions by MEAC treatment (**A**). The representative densitometry analyses of TNF-α (**B**), IL-6 (**C**), and IL-1β (**D**) compared with GAPDH mRNA. Data are presented as the mean ± SEM. ### *p* < 0.001 versus control group (*n* = 3); * *p* < 0.05 and *** *p* < 0.001 versus LPS group (*n* = 3).

**Figure 5 antioxidants-08-00348-f005:**
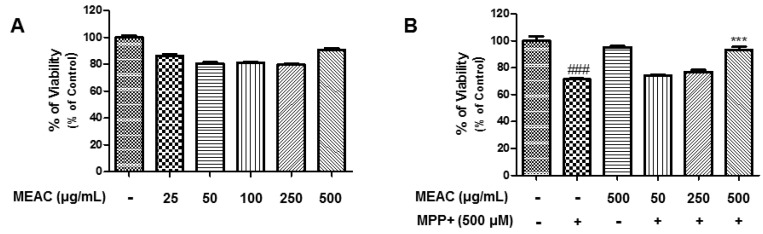
MEAC treatment on viability of N27-A cells. (**A**) Action of different doses of MEAC treatments on the viability of N27-A cells. (**B**) Action of MEAC treatment against 1-methyl-4-phenylpyridinium (MPP^+^)-induced death in N27-A cells. Data are presented as the mean ± SEM. ### *p* < 0.001 versus control group; *** *p* < 0.001 versus MPP^+^-induced group (*n* = 3).

**Figure 6 antioxidants-08-00348-f006:**
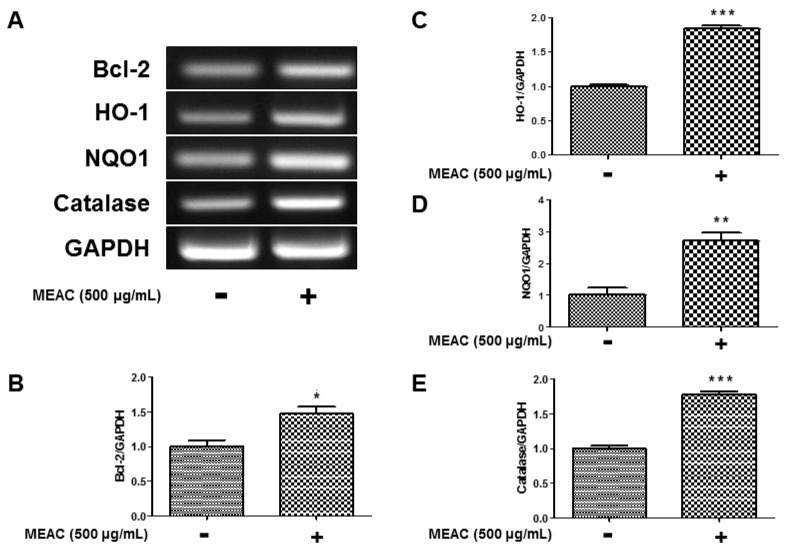
Effect of MEAC treatment on the expression of Bcl-2, HO-1, NQO1, and catalase at mRNA levels (**A**). The representative densitometry analyses of Bcl-2 (**B**), HO-1(**C**), NQO1 (**D**), catalase (**E**) compared with GAPDH mRNA. Data are presented as the mean ± SEM. * *p* < 0.05, ** *p* < 0.01 and *** *p* < 0.001 versus control group (*n* = 3).

**Table 1 antioxidants-08-00348-t001:** The list of PCR primers employed in the present study.

Gene Target.	Primer Sequence	Size (bp)
iNOS	F 5′-GAG GTA CTC AGC GTC CTC CA -3′ R 5′-AGG GAG GAA AGG GAG AGA GG-3′	444
COX-2	F 5′-TGA GTG GTA GCC AGC AAA GC-3′ R 5′-CTG CAG TCC AGG TTC AAT GG -3′	319
TNF-α	F 5′-TTC GAG TGA CAA GCC TGT AGC-3′ R 5′-AGA TTG ACC TCA GCG CTG AGT-3′	390
IL-1β	F 5′-CAA GGA GAA CCA AGC AAC GA-3′ R 5′-TTG GCC GAG GAC TAA GGA GT-3′	428
IL-6	F 5′-GGA GGC TTA ATT ACA CAT GTT-3′ R 5′-TGA TTT CAA GAT GAA TTG GAT-3′	435
Bcl-2	F-5′-CCA GGC CTT CAA CCA TTA TC-3′ R-5′-CTC ATT GAA CTC GTC TCC GA-3′	127
HO-1	F-5′ TGT CAC CCT GTG CTT GAC CT-3′ R-5′-ATA CCC GCT ACC TGG GTG AC-3′	200
NQO1	F-5′-AGA GCC CTG ATT GTA TTG GC-3′ R-AGG TCA GAT TCG ACC ACC TC-3’	113
Catalase	F-5′-CCT GAC ATG GTC TGG GAC TT-3′ R-5′-CAA GTT TTT GAT GCC GTG GT-3′	201
GAPDH	F 5′- ACC ACA GTC CAT GCC ATC AC -3′ R 5′- CCA CCA CCC TGT TGC TGT AG-3′	472

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
