# Peer review of "The Methanol Extract of Allium cepa L. Protects Inflammatory Markers in LPS-Induced BV-2 Microglial Cells and Upregulates the Antiapoptotic Gene and Antioxidant Enzymes in N27-A Cells"

_antioxidants, 2019, doi:10.3390/antiox8090348_

Round 1

Reviewer 1 Report

The manuscript “The methanol extract of Allium cepa L. protects inflammatory markers in LPS-induced BV-2 3 microglial cells and upregulates the antiapoptotic 4 gene and antioxidant enzymes in N27-A cells 5” by Md. Jakaria  et al. addresses an interesting question concerning the activity of the methanol extract of Allium cepa (MEAC) against neurodegenerative disease.

The article is well written and the results are clearly described. I have just some suggestions to improve the manuscript.

-          The authors should perform some tests to assess the plasma/serum stability of MEAC at body temperature.

-          Would MEAC be able to penetrate the human blood brain barrier? Could the authors address this question by experimental assays or/and add some comments about this question in the Discussion section of the manuscript?

Author Response

Dear Reviewer,

We are thankful to you for giving us your valuable time for reviewing our manuscript. Following your comments and suggestions, we have revised our manuscript. All changes are marked by green color. We hope your kind consideration on accepting our revised manuscript. Thanks again

Sincerely,

Prof. Dong-Kug Choi

Responses to reviewer’s comments

Review Report Form

Open Review

English language and style

( ) Extensive editing of English language and style required
( ) Moderate English changes required
(x) English language and style are fine/minor spell check required
( ) I don't feel qualified to judge about the English language and style

Yes

Can be improved

Must be improved

Not applicable

Does the introduction provide sufficient background and include all relevant references?

(x)

( )

( )

( )

Is the research design appropriate?

( )

(x)

( )

( )

Are the methods adequately described?

( )

(x)

( )

( )

Are the results clearly presented?

(x)

( )

( )

( )

Are the conclusions supported by the results?

(x)

( )

( )

( )

Comments and Suggestions for Authors

The manuscript “The methanol extract of Allium cepa L. protects inflammatory markers in LPS-induced BV-2 3 microglial cells and upregulates the antiapoptotic 4 gene and antioxidant enzymes in N27-A cells 5” by Md. Jakaria  et al. addresses an interesting question concerning the activity of the methanol extract of Allium cepa (MEAC) against neurodegenerative disease.

The article is well written and the results are clearly described. I have just some suggestions to improve the manuscript.

-          The authors should perform some tests to assess the plasma/serum stability of MEAC at body temperature.

Response: We thank you for bringing up this point. Nwaehujor et al. (2014) reported that with aqueous extract of A. cepa increases in osmotic fragility of red blood cells attributed to the disturbance in ionic motion through their membrane could be stabilized. Also, the extract ameliorates oxidation caused by reactive oxygen species and other free radicals produced in diabetic states. We are agreed with you it would be excellent if we can assess the plasma/serum stability of MEAC at body temperature. Due to some limitation on this project, we are unable to perform further experiments. However, we will consider your valuable comments in our future project.

-          Would MEAC be able to penetrate the human blood brain barrier? Could the authors address this question by experimental assays or/and add some comments about this question in the Discussion section of the manuscript?

Response: We are grateful to you for mentioning this comment on BBB. Hyun et al. (2013) addressed that A. cepa extract prevents brain edema, BBB hyperpermeability, and tight junction proteins disruption, possibly through its antioxidant effects in the mouse MCAO model. This study suggests that A. cepa may be a beneficial nutrient for the prevention of BBB function during brain ischemia. We have added this study in the discussion section of our manuscript (line 237-240).

Reviewer 2 Report

The paper entitled "The methanol extract of Allium cepa L. protects inflammatory markers in LPS-induced BV-2 microglial cells and upregulates the antiapoptotic gene and antioxidant enzymes in N27-A cells" by Jakaria et al. reports the anti-oxidant as well anti-inflammatory activity exerted by methanol extract of Allium Cepa.

The study was performed on rodent CNS cell lines since most of the neurodegenerative diseases recognize in inflammation and oxidative stress two important aspects of their pathogenetic process.

I think a few points need to be clarified further:

Why MEAC treatments were performed using different times? e.g. MEAC treatments of BV2 cells has been lasting for 1 hr before LPS addition (see Cox-2): Instead for the nitrite assay (pag.3) MEAC and LPS were added together; moreover it is not written if there was a MEAC pretreatment of BV2 cells to analyze the modulation of iNOS expression stimulated by LPS. There was a 2 hr MEAC pretreatment of N27-A cells before addition of MPP+.

Why is MAEC added to cultures in different ways from one experiment to another?

As far as the effect on cytokine expression is concerned, Authors showed only mRNAs modulation; I'd rather to see also some ELISA results, in order to see also the protein expression

I suggest to put together, in the same figure results about Bcl-2, HO-1, NQ01 and catalase

The text needs major revisions. Some sentences are hard to understand and there are also repetitions and typing errors (see lines 197,234, 84,243,245,251)

Author Response

Dear Reviewer,

We are thankful to you for giving us your valuable time for reviewing our manuscript. Following your comments and suggestions, we have revised our manuscript. All changes are marked by green color. We hope your kind consideration on accepting our revised manuscript. Thanks again

Sincerely,

Prof. Dong-Kug Choi

Review Report Form

Open Review

English language and style       

(x) Extensive editing of English language and style required
( ) Moderate English changes required
( ) English language and style are fine/minor spell check required
( ) I don't feel qualified to judge about the English language and style

Yes

Can be improved

Must be improved

Not applicable

Does the introduction provide sufficient background and include all relevant references?

( )

(x)

( )

( )

Is the research design appropriate?

(x)

( )

( )

( )

Are the methods adequately described?

( )

( )

(x)

( )

Are the results clearly presented?

( )

(x)

( )

( )

Are the conclusions supported by the results?

(x)

( )

( )

( )

Comments and Suggestions for Authors

The paper entitled "The methanol extract of Allium cepa L. protects inflammatory markers in LPS-induced BV-2 microglial cells and upregulates the antiapoptotic gene and antioxidant enzymes in N27-A cells" by Jakaria et al. reports the anti-oxidant as well anti-inflammatory activity exerted by methanol extract of Allium Cepa.

The study was performed on rodent CNS cell lines since most of the neurodegenerative diseases recognize in inflammation and oxidative stress two important aspects of their pathogenetic process.

I think a few points need to be clarified further:

Why MEAC treatments were performed using different times? e.g. MEAC treatments of BV2 cells has been lasting for 1 hr before LPS addition (see Cox-2): Instead for the nitrite assay (pag.3) MEAC and LPS were added together; moreover it is not written if there was a MEAC pretreatment of BV2 cells to analyze the modulation of iNOS expression stimulated by LPS. There was a 2 hr MEAC pretreatment of N27-A cells before addition of MPP+.

Responses: We are grateful to you for mentioning these comments. We had used two different cell lines such as microglial cell and dopaminergic cell lines. In addition, we have used toxins, including LPS and MPP+. According to the nature of the toxins and cell lines and previously published protocols, we have treated extract. However, we have pretreated extract on BV-2 microglial cell line in terms of RT-PCR and WB, but we treated cotreated extract and LPS in case of nitrite assay. As pretreatment or co treatment possible in terms of nitrite assay, we have chosen co treatment to assess the preliminary anti-inflammatory activity of extract.

Why is MAEC added to cultures in different ways from one experiment to another?

Response: Based on the cell lines and previously published protocol, MAEC added to cultures in different ways.

As far as the effect on cytokine expression is concerned, Authors showed only mRNAs modulation; I'd rather to see also some ELISA results, in order to see also the protein expression

Response: We thank you for mentioning this excellent comment. We are agreed with you it would be excellent if we can asses ELISA experiment. Due to some unavoidable circumstances, we are unable to perform further experiments in this project. However, we will consider your valuable comments on our future project.

I suggest to put together, in the same figure results about Bcl-2, HO-1, NQO1 and catalase

Response: Following your suggestion, we have putted Bcl-2, HO-1, NQO1 and catalase in one figure.

The text needs major revisions. Some sentences are hard to understand and there are also repetitions and typing errors (see lines 197,234, 84, 243,245,251)

Responses: We are grateful to you for bringing up this point. We have revised the English language in the entire manuscript (especially, 197, 234, 84, 243, 245, 251). In addition, we have edited our manuscript by professional native English speakers (Editing company: Cambridge proofreading).

Round 2

Reviewer 2 Report

I revised the new submission and even if the authors did not completely reply to my comments I think that the new version is improved and therefore I suggest to accept it for publication on Antioxidants

Author Response

Dear Reviewer,

We are thankful to you for giving us time to review our manuscript. Thanks again

Best Regards,

Prof. Dong-Kug Choi